# Construction and Characterization of Polyolefin Elastomer Blends with Chemically Modified Hydrocarbon Resin as a Photovoltaic Module Encapsulant

**DOI:** 10.3390/polym14214620

**Published:** 2022-10-31

**Authors:** Jin Hwan Park, Seok-Ho Hwang

**Affiliations:** Materials Chemistry and Engineering Laboratory, Department of Polymer Science and Engineering, Dankook University, Yongin 16890, Gyeonggi-do, Korea

**Keywords:** polyolefin elastomer, hydrocarbon resin, adhesion strength, encapsulant, photovoltaic module

## Abstract

In this study, polyolefin elastomer (POE) was blended with a chemically modified hydrocarbon resin (*m*-HCR), which was modified through a simple radical grafting reaction using *γ*-methacryloxypropyl trimethoxy silane (MTS) as an adhesion promotor to the glass surface, to design an adhesion-enhanced polyolefin encapsulant material for photovoltaic modules. Its chemical modification was confirmed by ^1^H and ^29^Si NMR and FT-IR. Interestingly, the POE blends with the *m*-HCR showed that the melting peak temperature (*T*_m_) was not changed. However, *T*_m_ shifted to lower values with increasing *m*-HCR content after crosslinking. Additionally, the mechanical properties did not significantly differ with increasing *m*-HCR content. Meanwhile, with increasing *m*-HCR content in the POE blend, the peel strength increased linearly without sacrificing their transmittance. The test photovoltaic modules comprising the crosslinked POE blend encapsulants showed little difference in the electrical performance after manufacturing. After 1000 h of damp-heat exposure, no significant power loss was observed.

## 1. Introduction

High-quality and long-durability component materials are very important for photovoltaic (PV) modules to operate for long lifetimes of more than 20 years while reducing the lifespan cost of solar cells [1,2,3,4,5]. Generally, crystalline silicon-based PV modules are composed of a top glass cover, a front-side polymeric encapsulant layer, crystalline silicon cells with metallization on the front and rear, a backside polymer encapsulant, and a polymeric backsheet [6]. Thus, the performance of PV module packaging in protecting solar cells from the outside environment directly affects the long-term reliability of PV modules. Particularly, the encapsulant layer as a packaging material of a PV module should serve to do high optical coupling, mechanical support, and electrical isolation of the solar cells and cell circuit components [7].

Thus far, ethylene-vinyl acetate copolymer (EVA) is widely applied as an encapsulant material for crystalline silicon-based solar cells because of its specific advantages such as high flexibility, good optical and mechanical properties, and field experience for more than 30 years [8,9,10]. Unfortunately, the weather resistance of EVA is poor, which decreases its transmittance and causes adhesion loss under ultraviolet (UV) light. Indeed, its main drawback is its ability to absorb water molecules, which makes it react with the acetate moiety of the EVA backbone to generate acetic acid [11]. Consequently, the generated acetic acid accelerates the PV module’s failure mechanism [12], including the corrosion of metal-based interconnections and cell metallization [13,14,15], shortening the lifetime of the PV module. Thus, these concerns necessitate the search for new polymers for PV module encapsulation. Besides EVA, polyvinyl butyral [16,17], ionomer resins [18,19], and silicone elastomer [20,21] have also been used, but they are still either very expensive for PV module applications or have concerns over long-term durability [11].

Recently, polyolefin elastomer (POE) has been considered an encapsulant for PV modules because of its high transmittance, persistent bonding, and good creep resistance with small deformation [7,22,23,24,25]. The thermoplastic POE copolymer consists of ethylene and 1-octene monomers, whose crystallization characteristics mainly depend on the content and spatial distribution of long-chain branches in the polymer backbone [26,27]. Although the aforementioned merits for POE are enough to apply the polymer encapsulant material for PV modules, its cohesion is still a minor drawback as POE is a saturated polyolefin material. Moreover, despite being on the market for several years, only a few publications are available describing POE’s adhesion improvement for lamination as well as its long-term reliability during the operation of PV modules.

Therefore, this work focused on developing an adhesion-enhanced PV encapsulant material instead of the classical EVA commonly used for crystalline silicon-based solar cells. For this purpose, we designed and synthesized a chemically modified hydrocarbon resin (*m*-HCR) using *γ*-methacryloxypropyl trimethoxy silane (MTS) to obtain both the tackifier characteristic and adhesion to glass substrates. Then, the expected polymer blend series, POE/*m*-HCR with the additives of dicumyl peroxide (DCP), were prepared with a secondary master batch process. Lastly, this work addressed the feasibility of a POE encapsulant material as a design solution for PV modules in crystalline silicon-based solar cells.

## 2. Materials and Methods

### 2.1. Materials

POE (ENGAGE^TM^ 8137; density: 0.864 g/cm^3^; MFI: 13 g/10 min) was purchased from the Dow Chemical Company (Midland, MI, USA). Hydrocarbon resin (HCR) (SUKOREZ^TM^ SU-625, hydrogenated DCPD resin; Mw = 425, Mn = 238, PDI = 1.786, Acid number 0.04 KOH mg/g) was generously donated by Kolon Industries (Seoul, Korea). DCP and MTS were purchased from Sigma-Aldrich Inc. (Milwaukee, WI, USA). Other organic solvents and chemicals were purchased from Samchun Chemicals Co. (Seoul, Korea). The solar cells used in this study were based on multicrystalline silicon (*p*-type, 200 µm thick, 13.5 × 14.0 cm in size) produced by Hae Sung Solar Co., Ltd. (Gimpo, Korea).

### 2.2. Synthesis of m-HCR

The grafting reaction on HCR was conducted thermally at a high temperature with a radical initiator. Twenty grams of HCR was dissolved in xylene (50 mL), and then 2 g of MTS and DCP (0.3 g) were added to the mixture. The mixture was stirred for 5 h at 120 °C under N_2_ gas. After the reaction, the mixture was cooled down to 25 °C and was poured into excess acetone to precipitate the product, which was filtered, washed with cold acetone several times, and then kept in vacuo at 60 °C for 1 day to dry completely (Figure 1). The MTS grafting content in *m*-HCR was about 2 mol-%.

### 2.3. Sample Preparation

Using a batch-type internal mixer (RheoComp system, MKE, Deajeon, Korea), neat POE and POE/*m*-HCR blend samples were mixed uniformly with DCP [by 2 phr (parts per hundred POE resin)]. The *m*-HCR contents in the POE blends were controlled as 3, 6, 9, and 12 phr, respectively. Melt compounding was conducted for 8 min at 105 °C and a rotation speed of 50 rpm. The uncrosslinked blend film was carefully prepared using a manual hot press under mild conditions (~110 °C), and the obtained film was cured in a manual hot press under 170 °C for 10 min for the instrumental analysis.

### 2.4. Determination of Gel Content

The gel contents were determined according to ASTM D2765-16. First, the crosslinked samples were placed in folded 120-mesh copper screen cages and, their weights were measured before immersion in xylene. The samples in the cage were extracted under refluxing for 6 h to remove uncrosslinked polymer parts from the blends. The residue specimens were dried at 120 °C to a constant weight and subsequently reweighted. The gel content was calculated using the following formula:(1)$$\mathrm{Gel}\;\mathrm{content}\;(\%)=\frac{w_{o}-w_{g}}{w_{0}}\times 100$$
where $w_{0}$ is the initial weight of the sample in the cage, and $w_{g}$ is the weight of the dried sample in the cage after xylene extraction.

### 2.5. Lamination Process for Preparing the PV Module

The encapsulant laminator curing adopted in this work used an automatic laminating machine (YDS-0707, Radiant Automation Equipment Co., Ltd.; Qinhuangdao, China) for 18-cell multicrystalline solar cell-based PV modules. The sample was first loaded to the laminator (with a temperature of 140 °C and a pressure of 100 kPa) and kept in there for 300 s. Then, the pressure was sequentially reduced to 90 kPa (30 s) and 55 kPa (30 s) on top of the samples. Thereafter, the pressure was reduced to 25 kPa and kept for 600 s and was finally vented to atmospheric pressure. The stack structure of the PV modules was backsheet/encapsulant/silicon solar cell/encapsulant/glass. The fabricated PV module and its stacking diagram are shown in Figure 2.

### 2.6. Equipment and Experiments

Nuclear magnetic resonance (NMR) for chemical structure analysis was performed on an Agilent 400 MHz NMR Magnet spectrometer (Agilent Technologies Inc., Santa Clara, CA, USA) using chloroform-*d*_1_ (CDCl_3_) as a solvent. Additionally, Fourier transform-infrared (FT-IR) spectra were recorded on a Nicolet iS20 spectrophotometer (Thermo Fisher Scientific, Waltham, MA, USA) in absorbance mode under air conditions. The thermal characterization was studied using a DSC 1 differential scanning calorimeter (Mettler Toledo Co., Greifensee, Switzerland). The scans were performed at a heating rate of 20 °C/min under a nitrogen gas atmosphere. A tensile test (ASTM D412) was performed using an Instron universal testing machine (model 6800; Instron Co., Norwood, MA, USA) equipped with a 1-kN load cell, and measurements were conducted at a constant crosshead speed of 500 mm/min. The tests were evaluated from the averages of at least five parallel tests. The peel strength for the POE—Glass interface was measured according to the ASTM 3330M method with a separation angle of 180° and a separation rate of 150 mm/min at 25 °C. For this test, glass (soda-lime glass) and a compound mode of POE blend film were laminated and cured together in the manual hot press under 150 °C at 0.1 MPa for 10 min. The characteristics of the encapsulated solar cell modules were measured using a Newport solar simulator (Newport Corp., Irvine, CA, USA). The current—Voltage (*I*–*V*) characteristics were determined using a digital source meter (Keithley 2400, Keithley Instruments, Inc., Solon, OH, USA) under standard testing conditions: irradiance of 1000 W/m^2^ with an AM 1.5 G spectrum at room temperature. The samples for the damp heat test were placed in the chamber at 85 °C and 85% relative humidity (RH) (85/85 test).

## 3. Results and Discussion

The *m*-HCR was obtained through a simple radical grafting reaction using MTS, which could play a role as an adhesion promotor to a glass surface. FT-IR measurements were performed to confirm the chemical structure of *m*-HCR, and the acquired spectra for the neat HCR and *m*-HCR are shown in Figure 1. The neat HCR displayed its characteristic vibration bands around 2800–2930 cm^−1^ attributed to the C–H stretching [28] of aliphatic HCR. After the chemical modification of HCR, the resultant *m*-HCR showed an obvious carbonyl vibration band at 1760 cm^−1^ besides the inherent vibration bands of HCR. This unique band indicated that MTS had been successfully grafted onto HCR.

^1^H and ^29^Si NMR measurements were performed to further confirm the grafting of MTS onto the HCR, and the results are shown in Figure 2. On the ^1^H NMR spectrum of *m*-HCR ((Appendix A)), an isolated chemical shift was detected at 3.56 ppm, which was assigned to the three methoxy (–OC*H*_3_) groups in the MTS moiety besides the expected signals for the HCR. Figure 2B shows the ^29^Si NMR spectrum for the *m*-HCR, and a new single resonance peak was detected at −42.5 ppm attributed to the silicon atom of the alkyl trimethoxysilane [C-*Si*(OCH_3_)_3_] group. From the results of the FT-IR, ^1^H NMR, and ^29^Si NMR spectra, we could conclude that the plausible grafting reaction of MTS on the HCR was performed successfully.

DSC thermograms recorded during the cooling and second heating scan for neat POE and its blends with *m*-HCR are presented in Table 1. In the case of the cooling scan for the uncrosslinked samples, the peak maximum (*T*_c_) of neat POE was detected at 37.0 °C, beginning at 27.3 °C and completing at 45.3 °C (Appendix A). The *T*_c_ of the POE blend samples decreased with increasing *m*-HCR content. However, such *T*_c_ difference was relatively small and less than 6 °C. On the second heating scan thermogram, the uncrosslinked neat POE showed one endothermic peak (*T*_m_) due to the melting of monoclinic crystals [29] measured at 64.5 °C. Furthermore, the *T*_m_ of the POE blend samples showed values similar to that of the neat POE, implying that the uncrosslinked POE blends have no significant impact on the second heating thermogram shape and shifting of the exothermic peak. However, the DSC thermograms of the crosslinked neat POE and their POE blend samples showed lower *T*_m_ values than those of the uncrosslinked neat POE and their POE blend samples because of the increased content of shorter crystallizable chain blocks induced by the crosslinking reaction [30,31]. As shown in Table 1, the *T*_m_ of the crosslinked POE blend samples increased slightly as the *m*-HCR content in the blends increased. This increase may be attributed to the reduced crosslinking density of the POE blend samples. Nonetheless, the *T*_c_ of the crosslinked samples was not changed. Meanwhile, the glass transition temperature (*T*_g_) of all samples increased with increasing *m*-HCR content because the *T*_g_ of *m*-HCR is higher than that of the other polyolefins [32]. As we can see in Appendix A (Appendix A), the observed single *T*_g_ for all samples indicated that the phase morphology of the POE blend samples with *m*-HCR was homogeneous.

The effects of *m*-HCR content on the gel content of the crosslinked neat POE and its blends are presented in Figure 3. According to a previous report [33], the silane coupling agent, MTS, in the POE curing system could accelerate the curing reaction, increasing the degree of crosslinking. In this study, the gel content of the crosslinked film samples slightly decreased with increasing *m*-HCR content. This decreasing trend was related to the decreasing degree of crosslinking. Therefore, the TMS grafting on the HCR should be a reasonable pathway to incorporating the coupling agent without any disturbance in the curing process.

The UV/visible (UV/vis) transmission spectra of the crosslinked neat POE and its blend samples are presented in Figure 4. On the basis of the lowest wavelength range on the UV/vis spectrum, the UV cut-off information for the material could be interpreted. The commercial encapsulant film containing a UV blocker shows a UV cut-off wavelength below 300–400 nm at a low transmittance of less than 10% [6]. The crosslinked neat POE and its blend samples blocked the radiation under 300 nm, having a transmittance of nearly 10%. However, the UV cut-off wavelength for the crosslinked neat POE slightly shifted to a lower wavelength than that of the crosslinked blend samples. This shift may have been caused by the presence of the acryl group of TMS in the blends [12]. The crosslinked neat POE transmitted 90% light effectively from 400 to 800 nm, whereas its crosslinked blend samples showed a similar transmittance. Indeed, this transmittance value is almost similar to that of EVA. This is one of the major advantages of the POE encapsulant over other encapsulant materials.

The tensile properties of the crosslinked neat POE and its POE blends with *m*-HCR are shown in Figure 5. In a previous study [34], the ultimate properties (tensile strength and elongation at breaks) of polypropylene blends with HCR depended strongly on the presence of the HCR, but the tensile modulus increased with increasing HCR content. Meanwhile, in the case of uncrosslinked POE blends with *m*-HCR, the ultimate properties, and tensile modulus values were similar to those of the uncrosslinked neat POE. These results indicated that the mechanical properties of the POE matrix could be sustained with the addition of *m*-HCR. With crosslinking, the tensile modulus and elongation at break values showed trends similar to those of the uncrosslinked POE blends. However, the tensile strength values slightly decreased with increasing *m*-HCR content because of the decreased degree of crosslinking of the POE blends.

The adhesion strength between the glass and the encapsulant film was measured with a universal testing machine using the 180° peel test according to ASTM 3330M. Peel adhesion strength is a very important factor in evaluating the bonding of different interfaces in the PV module to ensure its longevity. In commercial encapsulant materials, the profile of peel testing fluctuates because of the uneven peeling path between the encapsulant and glass surface [35]. Therefore, the estimated peel adhesion strength (Appendix A) was taken from the mean of five repetitive peel testing results [36]. As shown in Figure 6, compared with a standard EVA (adhesion strength, 80–100 N/cm) [6,36], the obtained average adhesion strength of the neat POE encapsulant showed an extremely low value of around 1.12 ± 0.46 N/cm at the glass—Encapsulant interface.

According to previous reports [10,37,38], free silane can improve the peel strength when it is grafted on the encapsulant polymer chain. As silane was already part of the *m*-HCR chain as grafted moiety in the POE blends with *m*-HCR, there was no need for additional grafting during lamination. With increasing *m*-HCR contents in the POE blends, the adhesion strengths increased linearly, indicating that the entire amount of the silane moiety added to the POE matrix promoted adhesion. These results showed that the improved adhesion strength can help prevent moisture diffusion and reduce the opportunities for interfacial debonding/delamination, hence improving the reliability of PV modules.

The damp-heat test is an accelerated test conducted at high levels of temperature and humidity and is an important test for screening applied materials to certify the longevity of PV modules for actual applications [39,40]. These accelerated conditions can affect the interfacial adhesion mechanism (e.g., mechanical interlocking, chemical bonding, molecular interfacial diffusion, and polar—Polar interaction). In this study, the thermal stability of the POE-based PV module under accelerated testing conditions was estimated using the damp-heat test (conducted at 85 °C and 85% RH for 1000 h as per the IEC-61215 recommendation). The typical *I*–*V* curves of the PV modules encapsulated by the POE blends with *m*-HCR are shown in Figure 7. From these curves, the fill factor and power conversion efficiency (PCE) values were calculated, and the results are presented in Table 2. The measured initial PCE values of the PV modules encapsulated by the neat POE or POE blends ranged from 19.2% to 20.7% because there was no transparency difference between the POE blend encapsulant and the neat POE. After 1000 h of the damp-heat test, the encapsulated PV modules did not show a significant power loss (less than 5%). Thus, the experimented encapsulant was considered thermally stable. From these results, it can be concluded that the POE blends with *m*-HCR are better candidates than the classical EVA encapsulant for module encapsulant applications.

## 4. Conclusions

In this study, a polyolefin encapsulant material designed for photovoltaic modules had been successfully constructed by compounding polyolefin elastomer (POE) with chemically modified hydrocarbon resin (*m*-HCR), which was prepared by radical reaction of MTS on the HCR. Using the MTS-grafted HCR (*m*-HCR), POE blend films were prepared by hot-pressing the blends after melt-mixing. The simple addition of *m*-HCR in the POE blends shifted their recrystallization temperature (*T*_c_) to lower temperatures. However, *T*_c_ did not change even with increasing *m*-HCR content after crosslinking. Indeed, the *T*_g_ of all samples increased with increasing *m*-HCR content and appeared as a single value, which indicated that the POE blend samples with *m*-HCR were homogeneous in the blends. Even with increasing *m*-HCR content in the POE blends, the transmittance values were almost similar to those of EVA (90%). The mechanical properties of the POE blends could be sustained even with the addition of *m*-HCR. After crosslinking, the tensile modulus and elongation at break values showed trends similar to those of the uncrosslinked POE blends, but the tensile strength values slightly decreased with increasing *m*-HCR content. Additionally, with increasing *m*-HCR content in the POE blend, the peel strength increased linearly. This means that the modified polyolefin elastomer can effectively be used to adhere to glass in PV modules. The damp-heat test results of the PV modules with the POE blends showed no significant power loss. This implies that the performance of PV modules encapsulated with the POE blends with *m*-HCR could be sustained. Regarding this result, it can be inferred that this POE blend system can play the role of weather-resistant encapsulant for PV modules. Finally, the POE blends with *m*-HCR could provide a feasible solution for the development of high-quality and long-durability component materials for PV modules.

## Data Availability

Not applicable.

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
