# Peer review of "Construction and Characterization of Polyolefin Elastomer Blends with Chemically Modified Hydrocarbon Resin as a Photovoltaic Module Encapsulant"

_polymers, 2022, doi:10.3390/polym14214620_

Round 1
Reviewer 1 Report
1.The title is very general and vague ,should be rendered more informative about the real paper content
2.page 2 lines 48-52 the term octylene and octyl do not correspond to the real molecular situation
3.The definite molecular structure of the HCR resins in term of MW, branching, MWD, unsaturation should be added.
4. Also after grafting MTS the grafting degree should be evaluated
5. The sense lines 170-172 about the effect of crosslinking on TM is absolutely unclear to me. Also it is not reported if also the crystallinity degree is affected.
6. It I surprising that the Tg does not change in uncrosslinked and crosslinked blends even after addition of the HCR moiety
7.The speculation in the first part of page 6,lines 185-192, is rather unclear and not scientifically sustainable. Analysing the crosslinking on melting point minor variation is not correct .
8.Also it is not clear how could MTS affects the process and which are the mechanism involved.
9.Line 201 at p.6: after grafting no acrylic groups remain but carbonyl groups may affect optical response
10.AS far as the UV protection is concerned it is surprising that the data are very similar with varied HCR –MTS content
11. Tha anlysis of mechanical properties simply show that surprisingly the additive hano effect.Why ?
12.The conclusion is just a summary of the work already described without new concepats and with the same points to be corrected as in the text.
Reviewer 2 Report
Hwang and coworker prepared adhesion enhanced polyolefin elastomer based encapsulant using a chemically modified hydrocarbon resin and studied for photovoltaic modules. The author tried to develop new polyolefin elastomer based material to replace the classical ethylene-vinyl acetate copolymer. The approach is interesting and cost-effective. The prepared sample are well characterized and new findings are well matched with the conclusion drawn from this study. Therefore, the manuscript can be accepted in Polymers with minor revision.
1) It would be difficult for the reader to follow the reaction scheme after reading the discussion and material and method. Therefore, it is recommended that insert one complete reaction scheme describing the synthesis of the sample and their content.
2) The designation used for the temperature are placed along with the value with space in whole manuscript. It should put one space in value and oC.
3) It would be good for the readers, if authors cite the following related articles into the introduction part: Polym. Chem., 2017, 8, 6416; R. Soc. open sci., 2018, 5, 180367
Round 2
Reviewer 1 Report
The authors provided appreciated attemptsto respond to all commnesnts.Some were adequately considered and appropriate modification reported in the revised ms.
Some few points stiill are not completely solved but the authros promise consideration in their future work.
I think the paper can be accepted in the new version